# The impact of the reform of rural collective property rights system on villagers' public participation: An empirical study based on CRRS 2020 data

Rui Li[1]*, Duanyang Gao[2]*

**1** College of Public Administration and Humanities, Dalian Maritime University, Dalian, China, **2** School of Social Research, Renmin University of China, Beijing, China

* rui.li@dlmu.edu.cn (RL); angela12321@163.com (DG)

**Data Availability Statement:** This study used the de-identifed data from the China Rural Revitalization Survey (CRRS) in 2020, which were conducted by Rural Development Institute, Chinese

## Abstract

The reform of rural collective property rights system is of great significance for protecting the collective asset rights and interests of villagers, activating rural resource elements, and achieving rural revitalization. This study is based on 284 village committee questionnaires and 7451 villager questionnaires from 10 provinces in China, and uses multi-layer linear regression models to explore the impact of the reform of rural collective property rights system on villagers' public participation. Research has found that:(1) the reform of rural collective property rights system that has been completed at the rural level can significantly enhance the public participation of villagers, including total participation ($\beta = 0.102$, $p \leq 0.1$), interest expression ($\beta = 0.044$, $p \leq 0.1$), and election voting ($\beta = 0.076$, $p \leq 0.05$). However, the impact on volunteer service is not significant ($\beta = -0.004$, $p > 0.1$); (2)mechanism analysis shows that the reform can reduce the probability of migration for work, and thus enhance the level of public participation; (3)heterogeneity analysis reveals that the reform has a greater impact on the villagers' public participation in suburban villages and villages with better collective economy; (4)further analysis discovers that villagers have heard that reform can significantly enhance their public participation. This study comprehensively explores the spillover effects of the reform at the village level on public participation at the individual level through the use of more accurate measurement indicators, empirical analysis methods, and large-scale authoritative data, providing strong support for proposing strategies to promote villages' public participation.

## 1. Introduction

Public participation, serving as a pivotal force in driving social progress and democratization, has garnered widespread acknowledgment for its significance [1,2]. Various types of public participation, such as voting, engaging in public decision-making, negotiating and discussing and providing volunteer services [3–5], not only effectively convey the needs of the public and

Academy of Social Sciences. These data are publicly available, and for research purposes only. Users need to apply in the name of an institution. Questionnaires and the datasets are available upon reasonable request and with permission of Rural Development Institute, Chinese Academy of Social Sciences. The datasets can be requested as: (a) visit to: http://rdi.cass.cn/dcsj/202306/t20230607_5643271.shtml; (b) fill out the data usage application form as website's required, and send it to the reserved email on the website (crrs_rdi@cass.org.cn or crrs_rdi@sina.cn). The staff will complete the review and reply to the email within 1 week.

**Funding:** R.L. received Grant from the Fundamental Research Funds for the Central Universities Grant (3132024324). The funders had no role in study design, data collection and analysis, decision to publish, or preparation of the manuscript.

**Competing interests:** The authors have declared that no competing interests exist.

ensure that policymaking is scientific and reasonable, but also stimulate the vitality and creativity of various sectors of society, thereby earning widespread attention and research. For instance, public participation can facilitate the sustainable development of the environment, local urban planning, the formulation of transportation policies, and numerous other fields. [6–10], while public apathy will be a critical barrier to public affairs [11]. However, despite the immense potential and value demonstrated by public participation in practice, its effectiveness faces challenges in some rural areas. For example, the old rural economy of farming in the United States presents conservative public participation [12]. Of course, this has also been confirmed by rural areas in developing countries such as Cameroon and Vietnam [13,14]. In rural China, rapid economic development and substantial migrant for work have led to profound changes in the social structure [15,16]. Families and individuals are gradually detaching from the village collective, tending towards individualization, leading to issues such as loose management of rural public affairs and indifference in interpersonal relationships [17]. These problems have further exacerbated the difficulty of democratic governance in rural society, posing unprecedented challenges. Therefore, how to actively guide villagers to participate in village public affairs has become particularly important. This will not only enhance the efficiency of public participation in rural China, but also have far-reaching implications for promoting comprehensive social progress in other countries facing similar issues.

The ongoing rural collective property rights system reform in China represents the fourth major reform practice and innovation after land reform, collectivization, and household contract responsibility system [18,19]. Since its implementation in 2014 [20], this reform has set the goal of "establishing a well-defined, fully empowered, smoothly transferable, and strictly protected rural collective property rights system with Chinese characteristics, and protecting and developing the legitimate rights and interests of villagers as members of rural collective economic organizations" [21]. By establishing a clear collective property rights system and operating collective assets in a market-oriented manner, a large amount of idle rural collective resource elements can be activated [22]. From 2014 to 2021, after five batches of pilot projects for this reform, the national inventory and verification of collective book assets amounted to 7.7 trillion yuan, including 3.5 trillion yuan of operating assets [23].

The overall progress of the reform has been smooth. Many scholars have explored from the perspective of economic benefits and pointed out that, from a micro perspective, the reform has enhanced the vitality of collective economic organizations, promoted collective income generation and increased villagers' income. From a macro perspective, the reform has boosted productivity and further activated new driving forces for rural development in China [19,24].

It is noteworthy that the governance effects of the reform of the rural collective property rights system have also begun to attract attention. This reform focuses on and revolves around the shareholding cooperative system, which entails quantifying and allocating collective operational assets to individuals and households on the basis of asset inventory and membership determination. Consequently, the reform has facilitated the reconstruction of power relations in villages, empowering villagers with more political and economic rights, enhancing their awareness and capacity to engage in rural public affairs, and enriching the effective forms of deliberative democracy in villages [25,26]. A study, utilizing survey data from 1657 villagers in 87 villages across 18 provinces in China in 2018, analyzes the impact of this reform on village democracy and find that the reform strengthens rural democracy and significantly promotes democratic governance in villages [27]. However, this study also has its limitations, as the independent variable of the reform of the collective property rights system is measured by villagers' responses to whether the reform has been completed.

Therefore, based on the actual situation in rural China, this study aims to more accurately measure the reform as well as villagers' public participation, in order to conduct better

empirical analysis. The findings of this study reveal that the reform of the rural collective property rights system has significantly promoted the public participation of villagers, especially in terms of total participation, interest expression, and election voting. However, its impact on volunteer service is limited. Mechanism analysis reveals that reform promotes villagers' public participation by reducing migration for work. Heterogeneity analysis emphasizes the reform has greater impact on suburban villages and villages with better collective economy. Further analysis indicates villagers have heard that reform can significantly increase their public participation. Suggestions of this study include further deepening the reform of rural collective property rights system, promoting local employment and entrepreneurship, implementing the reform differently based on actual village conditions and increasing the promotion and guidance efforts for the reform.

The contributions of this study are threefold. Firstly, use more accurate indicators to measure the reform of rural collective property rights system and public participation. For the reform of the rural collective property rights system, this study measures whether the reform has been completed based on direct question from village level questionnaires, while previous studies verified the completion of the reform through villagers' responses; for public participation, this study establishes three main indicators and one comprehensive indicator, namely interest expression type, election voting type, volunteer service type and total participation. In comparison, previous studies mostly used only one variable for measurement. Secondly, the multi-level linear regression model and other econometric analysis methods used in this study scientifically and comprehensively investigated the impact of the reform on villagers' public participation, as well as the mechanism and heterogeneity. Thirdly, in terms of data, this study used data from 284 village committee questionnaires and 7451 village questionnaires from 10 provinces in China, which is also authoritative and representative. Above of these contributions help proposing countermeasures and suggestions on how to better promote the public participation of villagers in the process of deepening reform.

## 2. Theoretical analysis and research hypothesis

The reform of the rural collective property rights system encompasses three key aspects: strengthen the management of rural collective assets, carry out property rights reform of operational assets, exploring the forms of realizing rural collective economy (Fig 1) [21]. Throughout the foundational work of this reform, the principle of respecting villagers' will is paramount. This principle entails harnessing villagers as the primary agents, fostering their innovation and creativity, granting them the right to choose, and ensuring their rights to information, participation, expression, and supervision. Ultimately, this transforms villagers into active participants and beneficiaries of the reform.

Consequently, the reform of the rural collective property rights system will affect villagers' public participation in three ways: firstly, by strengthening the management of rural collective assets through various measures, it establishes a foundation for villagers to engage in public participation. Experiences from other developing countries indicate that such approaches not only elevate villagers' awareness and trust in collective assets but also ignite their enthusiasm for participating in public affairs [28,29]. Secondly, promoting the rural collective management asset shareholding cooperative system guarantees villagers' shareholding rights as members of collective economic organizations [30]. This enables villagers to participate more directly in the decision-making and operations of the collective economy, thereby expanding the channels and depth of their public participation. Lastly, exploring effective collective economy models, encouraging villager innovation, and supporting the standardized transfers and transactions of rural property rights not only promote the diversified development of rural

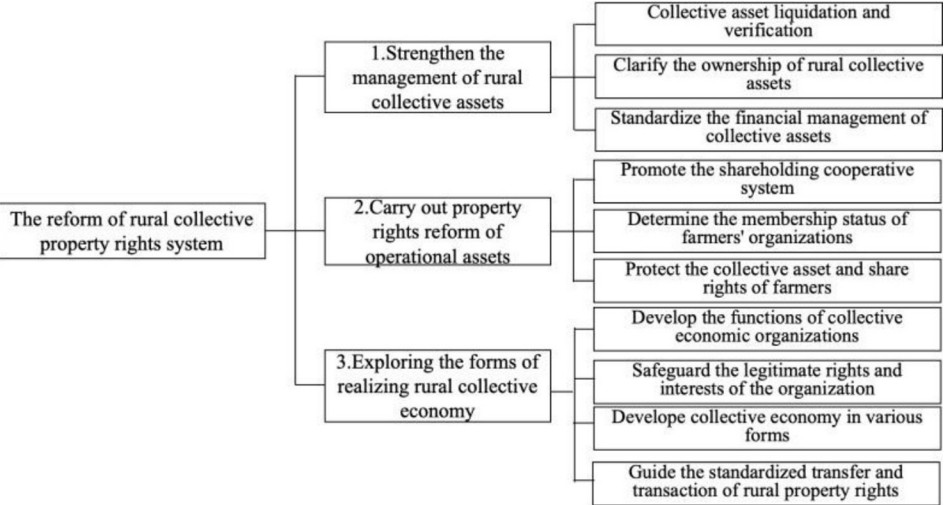

**Fig 1. The content of the reform of rural collective property rights system.**

economy, but also offer more platforms and opportunities for villagers' public participation [31,32]. By ensuring villagers' right to know, participate, express and supervise in the reform process, the transformation from villagers being mere observers to active participants and beneficiaries of the reform can be truly realized.

Based on the above analysis, this study presents research hypothesis 1:

Hypothesis 1(H1): The reform of rural collective property rights system can promote the villagers' public participation.

The positive reshaping of rural social structures through the rural collective property rights system cannot be overlooked. Research shows that the reform can effectively attract villagers who work outside to return home [33]. Behind this trend lies villagers' recognition and expectation of the development potential of their hometown, as well as their strong pursuit of personal rights and a sense of belonging. Returning villagers not only bring rich work experience and entrepreneurial enthusiasm, but also inject new vitality into rural communities, alleviating the issue of rural hollowing and providing a solid foundation for broader public participation among villagers [34,35]. In the process of public participation, villagers can not only express their own demands and interests more directly, but also jointly promote the harmony and development of rural society through collective action, achieving a win-win situation for both individuals and the collective.

Therefore, the hypothesis 2 can be proposed:

Hypothesis 2(H2): The reform of rural collective property rights system reduces the possibility of migration for work, thereby promoting villagers' public participation.

The impact of collective property rights system reform on villagers' public participation will be influenced by factors such as the geographical location and economic development level of the village. Firstly, based on the distance from the city center, rural areas can be divided into suburban villages and general villages [36]. Compared with general villages, suburban villages have their own development and governance characteristics. In terms of development, suburban villages have access to more resources and are more likely to take the lead in achieving development goals [37,38]. In terms of governance, suburban villages are often regarded as complex governance units with intertwined interests and mixed personnel, making them

more prone to conflicts [39]. But it has also received more attention and policy tilt. The reform of the collective property rights system emphasizes the need for suburban villages to actively establish and improve rural collective economic organizations, and promote the reform of the shareholding cooperative system for operating assets.

Similarly, previous studies have shown that well-developed collective economy not only provide a more solid material foundation and impetus for rural areas, but also greatly enhance villagers' profound understanding and strong sense of belonging towards the value of collective assets [40,41]. The deepening of this sense of belonging and cognition has laid a solid mass foundation for the smooth progress of rural collective property rights system reform. In this context, with the deepening of the reform of the collective property rights system, it is possible to more accurately and effectively clarify property rights relationships, effectively protect the legitimate rights and interests of villagers, and greatly stimulate their enthusiasm and initiative to public participation.

Thus, it is safe to propose the following hypotheses:

Hypothesis 3(H3): Compared to general villages, the reform of the collective property rights system in suburban villages has a more significant promoting effect on villagers' public participation.

Hypothesis 4(H4): The better the rural collective economy, the more obvious the promotion effect of the reform of the collective property rights system on villagers' public participation.

## 3. Research design

### 3.1 Data sources

The data used in this article comes from the China Rural Revitalization Survey (CRRS). CRRS is a large-scale national rural tracking survey initiated and conducted by Rural Development Institute, Chinese Academy of Social Sciences. The research team conducts a tracking survey every two years. The first survey of villagers and villages was conducted in 2020, surveying 50 county (city) units and 156 township (town) units in 10 provinces (regions) across the country. The research team took into account the level of economic development, regional location, and agricultural development, and randomly selected sample provinces from the eastern, central, western, and northeastern regions of China; using equidistant random sampling method to select sample counties based on the per capita GDP at the county level in the province, and considering covering the entire province (region) as much as possible in space; using the same sampling method, randomly select sample townships (towns) and sample villages based on the economic development level of local townships and villages; finally, sample households will be randomly selected based on the roster provided by the village committee. The sample provinces specifically include Guangdong Province, Zhejiang Province, Shandong Province, Anhui Province, Henan Province, Heilongjiang Province, Guizhou Province, Sichuan Province, Shaanxi Province, and Ningxia Hui Autonomous Region. A total of more than 15000 samples were obtained. The content covers personal basic information, family situation, health status and other information of villagers. After sample cleaning, 7451 village samples covering 284 villages were finally obtained.

### 3.2 Variable selection

**Dependent variable-public participation.** Referring to the previous studies [42,43], public participation can be divided into three categories: interest expression type, voting election type, and volunteer service type, and then select corresponding variables for measurement.(1)

Interest expression type. Sample selection is based on the questionnaire questions "Was a village assembly held in 2019?" and "How many times did you participate?". Assign a value of 1 if a villager assembly was held and the individual participated at least once. Assign a value of 0 if a villager assembly was held but the individual did not participate (i.e., participation count is 0). (2) Voting election type. Select "Did anyone in your family participate in the village committee election voting in 2019?" for measurement, and answer "Yes" with a value of 1 and "No" with a value of 0. (3) Volunteer service type. Measure based on the question "What is your reaction when a crisis occurs in the village?". The options include participating in the rescue at the first time, waiting for notification, contacting relatives or friends. The value for participating in the rescue at the first time is 1, otherwise it is 0. (4) In order to more comprehensively demonstrate the breadth of villagers' public participation, this study aggregates three types of public participation into a single score, ranging from 0 to 3 points.

Table 1 presents the definition and distribution of public participation in this study. It can be observed that villagers participate in, on average, more than two public participation types (Mean = 2.307). And the participation rates, from highest to lowest, are interest expression type, volunteer service type, and voting election type.

**Independent variable-the reform of rural collective property rights system.** Select the village survey questionnaire question "Has the reform of the rural collective property rights system been completed?". Assign a value of 1 if the answer is "Already Completed", and a value of 0 if the answer is "Not yet started" or "In progress". And then match the village survey data.

**Control variables.** Referring to the research on the influencing factors of public participation and combined with the CRRS questionnaire settings [44–47], this study controls for the following characteristic variables: the basic characteristics of rural areas include terrain, government distance, one shoulder multitasking, village secretary education; the basic characteristics of villagers include age, gender, education level, party, and income. In addition, in order to eliminate the influence of regional fixed effects, this study also controls for provincial-level dummy variables.

**Mediating variables.** The mediating variables in this study is migration for work. Based on the question "Did you and your family work outside of your hometown last year?", assigning a value of 1 to those who have worked outside their hometown and a value of 0 to those who have not.

The meaning, assignment, and descriptive statistics of the variables used in this study are shown in Table 2.

## 4. Empirical results and analysis

### 4.1 Benchmark regression results

Due to the fact that the reform of rural collective property rights system is conducted at the village level, while the villagers' public participation is at the individual level, in order to better

**Table 1. Basic frequency and percentage of public participation.**

| Type | Variables | Definition | Mean | S.D. | Min | Max |
|---|---|---|---|---|---|---|
| Total public participation | Aggregate public participation types | Number of public participation types | 2.307 | 0.652 | 0 | 3 |
| Interest expression type | Participate in village assembly | Participation = 1, otherwise = 0 | 0.867 | 0.339 | 0 | 1 |
| Election voting type | Participate in election voting | Someone from your family is voting in the village committee election = 1, otherwise = 0 | 0.463 | 0.499 | 0 | 1 |
| Volunteer service type | Participate in crisis rescue efforts | When a crisis occurs in the village, you participate in the rescue as soon as possible = 1, otherwise = 0 | 0.709 | 0.454 | 0 | 1 |

**Table 2. Descriptive statistics of variables.**

| Variables | Definition | Mean | S.D. | Min | Max |
|---|---|---|---|---|---|
| **Rural areas level** | | | | | |
| Reform | The reform of the rural collective property rights system been completed = 1, otherwise = 0 | 0.623 | 0.485 | 0 | 1 |
| Terrain | Plain = 1, non plain = 0 | 0.443 | 0.497 | 0 | 1 |
| Government distance | Distance between Village Committee and County Government (kilometre) | 24.163 | 17.658 | 1 | 125 |
| One Shoulder Multitasking | The village party organization secretary concurrently serves as the village committee director = 1, otherwise = 0 | 1.431 | 0.495 | 1 | 2 |
| Village secretary education | Primary school = 1, junior high school = 2, high school and vocational school = 3, college diploma and undergraduate = 4 | 3.187 | 0.832 | 1 | 4 |
| **Villager level** | | | | | |
| Age | Age of the interviewed villagers (years) | 40.319 | 21.208 | 0 | 99 |
| Gender | Male = 1, female = 0 | 0.517 | 0.500 | 0 | 1 |
| Education | Primary school = 1, junior high school = 2, high school or vocational high school = 3, college or university = 4 | 1.939 | 0.979 | 1 | 4 |
| Party | Member of the Communist Party of China = 1, otherwise = 0 | 0.121 | 0.326 | 0 | 1 |
| Income | Logarithmic total annual household income | 10.915 | 1.116 | 5.298 | 16.120 |
| Migration for work | You and your family work outside of your hometown last year? = 1, otherwise = 0 | 0.597 | 0.491 | 0 | 1 |

examine the impact of village-level factors on individual behavior, this study selects a multi-level linear regression model (HLM) in the benchmark regression. Specifically, if there are differences in the participation behavior of villagers between rural areas, it indicates that their behavior is influenced not only by individual factors, but also by village-level factors. In order to test whether there are significant inter-village differences in the participation of villagers, this study first conducts a null model test on the dependent variable.

The model is set as shown in Eqs (1) and (2):

$$\text{Layer 1:} \quad Participation_{ij} = \alpha_{0j} + \varepsilon_{ij} \tag{1}$$

$$\text{Layer 2:} \quad \alpha_{0j} = \beta_{00} + \mu_{0j} \tag{2}$$

Among them, i = 1, . . ., n represents the i-th villager; j = 1,. . .,j, indicates the j-th rural area. $Participation_{ij}$ as the dependent variable, representing the participation of the i-th villager in public participation in village j; $\alpha_{0j}$ represents the average value of public participation of villagers in the j-th village; $\beta_{00}$ represents the overall average value of villagers' public participation; $\varepsilon_{ij}$ represents the random error of Layer 1 and $\mu_{0j}$ represents the random error of Layer 2.

The results presented in Table 3 show that the Intra-class Correlation Coefficients (ICC(1)) for "Interest expression type", "Election voting type", and "Volunteer service type" are 0.229, 0.328, and 0.182, respectively. The ICC(1) for "Total participation" is 0.317. All these values exceed the commonly used threshold of 0.050. These indicate that the public participation behavior of villagers is influenced by factors at the village level, and it is appropriate to use a multi-layer linear regression model for empirical analysis.

After conducting null model testing, this study proceeds to test hypothesis 1 by setting up the following multilevel linear regression model:

$$\text{Layer 1:} \quad Y_{ij} = \alpha_{0j} + X_{ij} + \varepsilon_{ij} \tag{3}$$

$$\text{Layer 2:} \quad \alpha_{0j} = \beta_{00} + \beta_{01} \ Reform_j + \xi_j + \mu_{0j} \tag{4}$$

**Table 3. Null model regression results.**

| Variable | Total public participation | Interest expression type | Election voting type | Volunteer service type |
|---|---|---|---|---|
| ICC(1) | 0.317 | 0.229 | 0.328 | 0.182 |
| LR test-chi2 | 2046.110*** | 1288.49*** | 2265.34*** | 825.72*** |
| Constant | -0.610*** | -1.205*** | -0.893*** | -0.878*** |
|  | (-0.008) | (-0.008) | (-0.008) | (-0.008) |
| N(L1) | 7451 | 7451 | 7451 | 7451 |
| N(L2) | 284 | 284 | 284 | 284 |

Note: Standard errors in parentheses

*** p < 0.01.

Among them, $X_{ij}$ represents the control variable matrix of individual characteristics, $Reform_j$ is the independent variable of the reform of rural collective property rights system, and $\xi_j$ represents the control variable matrix of village characteristics. $\beta_{01}$ represents the degree to which the reform affects the villagers' public participation, and $\xi_{0j}$ is the coefficient that this study focuses on. The meanings of other variables and letters are the same as above.

As shown in Table 4, it can be observed that after controlling for possible related variables, the reform of rural collective property rights system has a significant positive impact on villagers' total public participation ($\beta = 0.102$, $p \leq 0.1$), as well as on interest expression ($\beta = 0.044$, $p \leq 0.1$) and election voting ($\beta = 0.076$, $p \leq 0.05$). However, the impact of the reform on villagers' participate in volunteer service is not significant. Overall, the results of the benchmark regression indicate that the reform of rural collective property rights system is conducive to promoting villagers' public participation, and hypothesis 1 has been validated.

The aforementioned results are primarily attributed to the economic incentives stemming from clear property rights and the enhancement of rights awareness [48], as mentioned in the analysis of H1. Due to the reform of rural collective property rights system, villagers have become increasingly attentive to the distribution of collective benefits, which has promoted an increase in the overall public participation, manifested as broader willingness and actions to participate. Among them, the notable increase in participation in interest expression and election voting may be due to the reform providing villagers with clearer channels for their interests and a fairer election environment, thereby enhancing their willingness and ability to express their opinions and exercise their rights.

However, the participation in volunteer service has not increased significantly, which may be because volunteer services are mostly based on individual voluntarism and dedication [49]. Although the reform has enhanced villagers' awareness of economic rights, it may not directly translate into motivation for voluntary service. Furthermore, volunteer service requires a certain degree of organizational mobilization and social atmosphere [50], and during the reform process, insufficient attention may have been given to these non-economic incentives. In summary, future reform should comprehensively consider both economic incentives and sociocultural factors, promoting villagers' comprehensive public participation from multiple dimensions.

## 4.2 Robustness test

**1. Replace the benchmark regression.** As the participation of villagers in a certain type of public participation is a binary variable, the multi-layer linear regression model in the benchmark regression is replaced with a Probit model for robustness testing. The regression results,

**Table 4. Benchmark regression results.**

| Variable | (1) | (2) | (3) | (4) |
|---|---|---|---|---|
| | Total public participation | Interest expression type | Election voting type | Volunteer service type |
| Reform | 0.102* | 0.044* | 0.076** | -0.004 |
| | (0.054) | (0.026) | (0.037) | (0.030) |
| Terrain | 0.089* | -0.024 | 0.075** | 0.014 |
| | (0.053) | (0.026) | (0.037) | (0.030) |
| Government distance | -0.000 | -0.001 | 0.001 | -0.001 |
| | (0.001) | (0.001) | (0.001) | (0.001) |
| One Shoulder Multitasking | -0.028 | 0.058* | -0.122*** | -0.010 |
| | (0.061) | (0.029) | (0.035) | (0.034) |
| Village secretary education | 0.028 | 0.002 | 0.028 | -0.006 |
| | (0.027) | (0.013) | (0.019) | (0.016) |
| Age | 0.001* | 0.000 | 0.000 | -0.000* |
| | (0.000) | (0.000) | (0.000) | (0.000) |
| Gender | -0.003 | -0.006 | 0.002 | -0.006 |
| | (0.013) | (0.007) | (0.010) | (0.010) |
| Education | 0.012* | 0.009** | 0.006 | 0.004 |
| | (0.007) | (0.004) | (0.005) | (0.005) |
| Party | -0.001 | 0.051*** | -0.049*** | 0.061*** |
| | (0.021) | (0.012) | (0.016) | (0.016) |
| Income | -0.004 | 0.005 | -0.015*** | 0.037*** |
| | (0.007) | (0.004) | (0.005) | (0.005) |
| Province | Yes | Yes | Yes | Yes |
| Constant | 2.076*** | 0.733*** | 0.486*** | 0.562*** |
| | (0.171) | (0.085) | (0.111) | (0.105) |
| N(L1) | 7451 | 7451 | 7451 | 7451 |
| N(L2) | 284 | 284 | 284 | 284 |

Note: Standard errors in parentheses

* $p < 0.1$

** $p < 0.05$

*** $p < 0.01$.

as shown in Table 5, reveal that they are consistent with the benchmark regression results, indicating that the benchmark regression results are relatively robust.

**2. Adjusting regional fixed effects.** In the benchmark regression, in order to eliminate the influence of regional fixed effects, provincial-effect were also controlled. Further control variables for county, town, and regional level dummy variables are included in Table 6. It can be seen that the regression results still indicate that the reform of rural collective property rights system can significantly improve the level of public participation of villagers.

### 4.3 Mechanism analysis

In order to investigate in depth the channels through which the reform of the collective property rights system affects the villagers' public participation, this study constructs a mediation effect model to test research hypothesis 2. Firstly, examine the effect of the independent variable on the dependent variable, then examine the effect of the independent variable on the mediator variable, and finally examine the combined effect of the independent and mediator variables on the dependent variable. The analysis results are shown in Table 7.

**Table 5. Robustness test based on probit model.**

| Variable | (1) | (2) | (3) | (4) |
|---|---|---|---|---|
| | Total public participation | Interest expression type | Election voting type | Volunteer service type |
| Reform | 0.166*** | 0.219*** | 0.138*** | -0.018 |
| | (0.032) | (0.044) | (0.036) | (0.037) |
| Control variables | Yes | Yes | Yes | Yes |
| Chi$^2$ | 529.745 | 135.468 | 869.609 | 312.178 |
| R$^2$ | 0.037 | 0.023 | 0.085 | 0.035 |
| N | 7451 | 7451 | 7451 | 7451 |

Note: Standard errors in parentheses

* p < 0.1

** p < 0.05

*** p < 0.01.

Column (1) is the same as the first column in Table 4, indicating that the reform of the collective property rights system improves the villagers' public participation (β = 0 102, p ≤ 0.1). Column (2) shows that the reform significantly reduces the probability of villagers migration for work (β = -0) 062, p ≤ 0.1). Column (3) suggests that including migration for work in the regression equation of the Column (1) still enhances the level of villages' public participation (β = 0 104, p ≤ 0.1). This indicates that migration for work plays a mediating role in the influence of the reform of the collective property rights system on villagers' public participation, and verifies hypothesis 2.

Reform often accompanies the development and growth of collective economy, providing villagers with more employment opportunities and sources of income [51]. This has to some extent reduced the willingness and probability of migration for work, enabling more villagers to stay locally and have more time and energy to invest in village governance. Retaining villagers not only enhances the village's human capital but also provides more opportunities for organizing and implementing public participation activities. In summary, the reform of the rural collective property rights system effectively elevates villagers' public participation by reducing the probability of them migration for work.

**Table 6. Robustness test based on adjusting regional fixed effects.**

| Variable | (1) | (2) | (3) |
|---|---|---|---|
| Reform | 0.114* | 0.182** | 0.100** |
| | (0.063) | (0.087) | (0.050) |
| Control variables | Yes | Yes | Yes |
| County level effect | Yes | | |
| Town level effect | | Yes | |
| Regional effect | | | Yes |
| Constant | 1.628*** | 2.635*** | 2.146*** |
| | (0.261) | (0.481) | (0.149) |
| N | 7451 | 7451 | 7451 |

Note: Standard errors in parentheses

* p < 0.1

** p < 0.05

*** p < 0.01.

**Table 7. Mechanism analysis.**

| Variable | (1) | (2) | (3) |
|---|---|---|---|
| | **Total public participation** | **Migration for work** | **Total public participation** |
| Reform | 0.102* | -0.062* | 0.104* |
| | (0.054) | (0.034) | (0.054) |
| Migration for work | | | 0.035** |
| | | | (0.014) |
| Control variables | Yes | Yes | Yes |
| Constant | 2.076*** | 0.185 | 2.069*** |
| | (0.171) | (0.115) | (0.171) |
| N | 7451 | 7451 | 7451 |

Note: Standard errors in parentheses

* p < 0.1

** p < 0.05

*** p < 0.01.

## 4.4 Heterogeneity analysis

The previous analysis verified the impact of the reform of rural collective property rights system on villagers' public participation. However, the social situation of different villages will have varying impacts on the effectiveness of the reform. Therefore, this study uses whether rural areas are located in the suburban areas as a proxy variable for rural geographical location, and uses the rural collective economy as a proxy variable for the rural economy to further conduct heterogeneity analysis.

Table 8 presents the results of heterogeneity analysis. From the perspective of urban proximity(first two columns), compared to general villages($\beta = 0.075$, p>0.1), the reform has a greater impact on the villagers' public participation in suburban villages($\beta = 0.135$, p≤0.05). Suburban villages are more susceptible to urban radiation. Study has shown that urban residents have a higher willingness in public participation [52]. Therefore, for rural areas located around cities, the enthusiasm of villagers' public participation after the reform of the collective property rights system is further stimulated.

**Table 8. Heterogeneity analysis.**

| Variable | (1) | (2) | (3) | (4) |
|---|---|---|---|---|
| | **Suburban villages** | | **Rural collective economy** | |
| | **No** | **Yes** | **Less than or equal to the average level** | **Greater than average level** |
| Reform | 0.075 | 0.135** | 0.074 | 0.231** |
| | (0.130) | (0.059) | (0.064) | (0.114) |
| Control variables | Yes | Yes | Yes | Yes |
| Constant | 1.466*** | 2.117*** | 1.994*** | 2.000*** |
| | (0.558) | (0.184) | (0.195) | (0.392) |
| N | 1660 | 5791 | 5599 | 1852 |

Note: Standard errors in parentheses

* p < 0.1

** p < 0.05

*** p < 0.01.

From the perspective of the rural collective economy (last two columns), compared to villages with a collective economic development level of less than or equal to that of the average ($\beta = 0.074$, $p > 0.1$), the reform of the collective property rights system has a greater impact on the villagers' public participation with a collective economic development level greater than that of the average ($\beta = 0.231$, $p \leq 0.05$). Therefore, the effect of stimulating villagers' public participation through reform is more significant in villages with better economic development levels [53]. Therefore, hypotheses 3 and 4 have also been verified to be correct.

## 4.5 Further analysis

Finally, a significant endeavor of this study lies in the fact that, in previous research, whether the reform of the collective property rights system was completed was mostly confirmed through villagers' responses. Therefore, in this section, the independent variable in the benchmark regression, which was originally based on official answers indicating whether the village had completed the reform, is replaced with individual villagers' responses, to further analyze the scientific validity of this substitution. Table 9 presents further analysis results.

Column (1) presents the regression results for the entire sample based on the question, "Have you heard of the reform of the collective property rights system", which serves as a proxy variable for the reform. The regression results indicate that having heard of the reform significantly enhances villagers' total public participation ($\beta = 0.087$, $p \leq 0.01$).

Column (2) shows the regression results for the subset of samples where the village-level reform of the collective property rights system has been completed (i.e., Reform completed = 1), using the same question as in Column (1). The results again demonstrate that having heard of the reform significantly boosts villagers' total public participation ($\beta = 0.081$, $p \leq 0.01$).

Column (3), however, presents the regression results based on the question, "Do you know the rural collective property rights system reform is currently underway (or has already been completed) in this village?". Surprisingly, the results indicate that villagers who have heard of the village-specific reform do not have a significant impact on their total public participation, contrary to the previous conclusions [27].

These discoveries are very interesting and of great significance. On the one hand, it emphasizes the importance of information transparency and policy promotion [54]. When villagers fully understand the reform policies, they are better able to clarify their own rights and

**Table 9. Further analysis.**

| Variable | (1) | (2) | (3) |
|---|---|---|---|
| | **Total sample** | **Reform villages sample** | |
| Heard of the reform | 0.087*** | 0.081*** | |
| | (0.015) | (0.019) | |
| Heard of the reform is currently undergoing in village | | | 0.019 |
| | | | (0.020) |
| Control variables | Yes | Yes | Yes |
| Constant | 2.147*** | 2.068*** | 2.235*** |
| | (0.167) | (0.201) | (0.200) |
| N | 7398 | 4616 | 4173 |

Note: Standard errors in parentheses

\* $p < 0.1$

\*\* $p < 0.05$

\*\*\* $p < 0.01$.

responsibilities, actively express their interests and demands, and participate in social affairs [55]. On the other hand, the previous use of villagers' awareness of reforms as a basis for studying the effectiveness of collective property rights system reform needs to be approached with caution and deserves further discussion.

## 5. Conclusions and policy implications

### 5.1 Conclusions

Based on the data from the China Rural Revitalization Survey (CRRS) 2020, which collected 284 village committee questionnaires and 7451 villager questionnaires from 10 provinces in China, this study explores the impact of the reform of rural collective property rights system on villagers' public participation and obtain the following main findings.

First, the analysis results based on a multi-layer linear regression model indicate that the completed rural collective property rights system reform can significantly promote the villagers' public participation. Specifically, the reform of the rural collective property rights system has respectively increased the total public participation ($\beta = 0.102$, $p \leq 0.1$), interest expression public participation ($\beta = 0.044$, $p \leq 0.1$), and election voting public participation ($\beta = 0.076$, $p \leq 0.05$), and the coefficient of overall public participation breadth is the highest, and the significance of election voting public participation is the strongest. However, the impact of the rural collective property rights system reform on villagers' voluntary service public participation is not significant ($\beta = -0.004$, $p > 0.1$). After that, this study mainly used the total participation breadth for robustness testing, and the results are still stable.

Second, mechanism analysis reveals that the reform of rural collective property rights system can improve the participation level of villagers' public participation by reducing the probability of migration for work. This reveals the path for the reform to enhance the level of villagers' public participation.

Third, the results of heterogeneity analysis indicate that the impact of the reform on villagers' public participation varies according to the geographical location and collective economy. Specifically, in terms of geographical location, suburban villages can significantly boost villagers' enthusiasm for participating in social governance after the reform. In terms of collective economy, villagers in villages with a relatively well-developed economies have stronger motivation to engage in social governance following the implementation of the reform.

Fourth, further analysis also found that villagers' awareness of the reform of the collective property rights system has an enhancing effect on their public participation, but their knowledge of the reform in their village has no effect on their public participation.

### 5.2 Policy implications

Based on the above conclusions, this study proposes the following policy recommendations. Firstly, further deepen the reform of rural collective property rights system. The core finding of this study is that the reform of rural collective property rights system serves as a crucial prerequisite and powerful impetus for promoting villagers' public participation. After years of reform efforts, China has established a relatively mature reform approach. What makes this conclusion even more practically significant is the promulgation of the "Law of the People's Republic of China on Rural Collective Economic Organizations" in June 2024, which responds to and solves many questions arising during the reform process from a institutional perspective and will officially take effect in January 2025 [56]. However, according to existing surveys, the coverage of the reform in China still needs further improvement [57]. Therefore, leveraging the existing legal support, it is imperative to persist in and deepen this reform direction, continually enhancing the property rights system and clearly delineating villagers' rights and

interests in collective assets, thereby further stimulating their enthusiasm and motivation in public participation.

Secondly, promote local employment and entrepreneurship. The mechanism by which the reform of rural collective property rights system enhances villagers' public participation provides an important direction for policymakers: to create more employment and entrepreneurial opportunities for villagers. While advancing the reform, it is necessary to intensify support for rural collective economy. This involves not only providing more financial support but also introducing preferential policies to incentivize and assist rural collective economic organizations in strengthening their economic capabilities. Specific measures include developing industries with distinctive local characteristics, optimizing existing industrial structures, and introducing advanced science and technology [58]. Of course, the government also needs to increase investment in rural infrastructure construction and public service systems, continuously improving rural production and living environments, in order to attract migrant workers to return home for employment or self-employment. As a result, villagers' willingness to migration for work will decrease, enabling them to enjoy the benefits of the rural collective property rights reform while having more time and energy to participate in public affairs, thereby promoting the overall level of public participation.

Thirdly, implement the reform differently based on actual village conditions. In the process of formulating and implementing reform, full consideration must be given to the differences in geographical location and economic development levels. For villages located in the suburbs of cities and those with higher levels of economic development, reforms should be further deepened and expanded, fully leveraging their demonstration and leading roles in enhancing villagers' public participation. Meanwhile, for villages in relatively remote geographical locations and those with lower levels of economic development, they should be encouraged to closely integrate with their own realities and actively explore reform paths and governance models that suit their unique characteristics, aiming to achieve precision and effectiveness in reform and development, and promote comprehensive rural revitalization.

Fourthly, increase the promotion and guidance efforts for the reform. Government should intensify its efforts in promoting policies concerning the reform of the rural collective property rights system and emphasize the transparency of information dissemination. Through a variety of channels and methods, including convening villager assemblies for detailed explanations, distributing promotional brochures, and publishing interpretations on online platforms, the government should comprehensively disseminate information about the reform policies, objectives, and their significance to villagers. The objective is not only to raise villagers' awareness of the reform but also to ensure they have a deep understanding of the tangible benefits it can bring. Additionally, the government needs to strengthen policy guidance, encouraging villagers to actively participate in social governance activities, particularly volunteer service activities, in order to promote comprehensiveness and diversity in public participation. Furthermore, it is necessary to explore effective incentive and safeguard mechanisms to ensure that villagers can genuinely engage in and benefit from the reform and governance process.

### 5.3 Limitations and future research directions

The limitations of this study include the following aspects. Firstly, in terms of research methodology. The reform of the rural collective property rights system commenced in 2014 and was piloted in batches. Therefore, the Difference-in-Differences (DID) method serves as an ideal empirical strategy to assess the effectiveness of the reform, be it in terms of economic or social effects. However, due to the lack of publicly available continuous survey data from CRRS, this study only uses linear regression methods based on cross-sectional data for

empirical analysis. Future studies should consider employing panel data and optimizing the methodology. Secondly, in terms of indicators of public participation, this study measures the types of public participation through three dimensions, but only one proxy variable is used for each dimension. In the future, more proxy variables should be considered to make the research conclusions more robust. Thirdly, in terms of research content, quantitative methods adopted in this study demonstrates positive social effects of the reform. Future research should also incorporate more mediating variables to delve deeper into the mechanisms underlying the relationship between the reform and public participation.

## Author Contributions

**Conceptualization:** Rui Li.

**Data curation:** Rui Li.

**Formal analysis:** Rui Li, Duanyang Gao.

**Funding acquisition:** Rui Li.

**Project administration:** Rui Li.

**Supervision:** Rui Li, Duanyang Gao.

**Writing – original draft:** Rui Li.

**Writing – review & editing:** Rui Li, Duanyang Gao.

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
