## [Decision Letter · Decision Letter 0]

20 Nov 2024

PONE-D-24-33936The impact of the reform of rural collective property rights system on villagers' public participation: An empirical study based on CRRS 2020 dataPLOS ONE

Dear Dr. Li,

Thank you for submitting your manuscript to PLOS ONE. After careful consideration, we feel that it has merit but does not fully meet PLOS ONE’s publication criteria as it currently stands. Therefore, we invite you to submit a revised version of the manuscript that addresses the points raised during the review process.

We look forward to receiving your revised manuscript.

Kind regards,

Dingde Xu

Academic Editor

PLOS ONE

Journal Requirements:

3. For studies involving third-party data, we encourage authors to share any data specific to their analyses that they can legally distribute. PLOS recognizes, however, that authors may be using third-party data they do not have the rights to share. When third-party data cannot be publicly shared, authors must provide all information necessary for interested researchers to apply to gain access to the data. (https://journals.plos.org/plosone/s/data-availability#loc-acceptable-data-access-restrictions)

Reviewers' comments:

Reviewer's Responses to Questions

**Comments to the Author**

1. Is the manuscript technically sound, and do the data support the conclusions?

Reviewer #1: Yes

Reviewer #2: Yes

2. Has the statistical analysis been performed appropriately and rigorously? 

Reviewer #1: Yes

Reviewer #2: Yes

3. Have the authors made all data underlying the findings in their manuscript fully available?

Reviewer #1: Yes

Reviewer #2: Yes

4. Is the manuscript presented in an intelligible fashion and written in standard English?

Reviewer #1: No

Reviewer #2: Yes

5. Review Comments to the Author

Reviewer #1: Dear authors:

Based on CRRS 2020 data, this paper analyzes the impact of rural collective property rights system reform on villagers' public participation, the research topic is meaningful. In general, the data source of the article is clear, the method is relatively scientific, the analysis is appropriate, and the logic is clear. Several revision suggestions are put forward for the author's reference:

1 The first paragraph of this article is not well written, it is too "Chinese". As an international magazine, it is very important to propose the research topic of this article from an international perspective, but this article does not do this.

2 The literature review and research hypothesis of this article are put together, and this arrangement does not seem to be logical. Generally, the literature review is carried out in the introduction part, or the literature review is delayed. The hypothesis is put together with the theoretical mechanism, and the second paragraph of the current article should be more appropriate as a theoretical mechanism and research hypothesis.

3 In the empirical part, the current empirical part lacks theoretical depth. It is recommended to combine the hypotheses in the previous article for a more in-depth analysis to increase the theoretical depth.

4 Although the policy implications are mentioned in the last part of this article, they are too weak and the content is too little. It is recommended to propose more targeted policies based on the quantitative results to highlight the practical significance of this article's research.

5 In terms of language, the language of this article currently has grammatical errors and "Chinese English" phenomena. It is recommended to reorganize the language.

I hope the above suggestions can contribute to your article!

Reviewer #2: Based on 284 village committee questionnaires and 7,451 villagers' questionnaires from 10 provinces in China, this study utilizes multilayer linear regression modeling to explore the impact of the reform of the rural collective property rights system on villagers' public participation. For better publication of this paper, the authors may wish to refer to the following comments.

1.The summary of the paper is clear and reasonable, although the authors might consider refining the last section, e.g., it might be better to describe the contributions of the paper rather than to list policy implications.

2.The format used by the author is not conducive to the reading of the article, and it is recommended that the author use a two-sided aligned layout, which may be more helpful.

3.The content of the introduction section is reasonable as a whole, but the lack of an author's summary of the paper's findings and a statement of the value of the paper's research in the introductory section may be incomplete.

4.The authors present the hypothesis of this paper in the second part, therefore the title is inappropriate, perhaps “Literature Review and Hypothesis”would be more appropriate.

5.The three-line table format is not standardized, and it is recommended that authors use a standardized three-line table for the presentation of tables. And each table is different, it is recommended that the standardized three-line table be used consistently.

6.The section on policy recommendations is too long, and it is recommended that the authors appropriately segment the content of the article, which would be more helpful for readers to understand the study.

7.The “5. Discussion” section is overloaded with conclusions, policy implications and discussion, of which the conclusions and policy implications should be placed in “6. Conclusions and policy implications”, whereas the current “6. Conclusion” might be more appropriate in the introduction. Conclusions and policy implications should be placed in “6. Conclusion and policy implications”, whereas the current “6. Conclusion” might be more appropriately placed in the introduction. It is recommended that the authors re-frame the content of the article and use the correct title to match the content.

6. PLOS authors have the option to publish the peer review history of their article (what does this mean?). If published, this will include your full peer review and any attached files.

Reviewer #1: **Yes: **Anlu Zhang

Reviewer #2: **Yes: **Hui Xiao

---

## [Author Response · Author response to Decision Letter 0]

27 Nov 2024

Title: The impact of the reform of rural collective property rights system on villagers' public participation: An empirical study based on CRRS 2020 data (Manuscript ID: PONE-D-24-33936)

Point-by-point response to reviewers

Reviewer #1:

Comment 1: The first paragraph of this article is not well written, it is too "Chinese". As an international magazine, it is very important to propose the research topic of this article from an international perspective, but this article does not do this.

Reply: Thanks for the constructive comments and we are gladly take the suggestion. Based on your suggestions, we have reconsidered and written the Introduction section, with the main modifications as follows:

(1)The first paragraph focuses on public participation, emphasizing its significance and the widespread attention given to it in existing research. However, it also points out that the effectiveness of public participation in rural areas of some countries, such as the United States, Cameroon, Vietnam, and China, which is the focus of this study, is still unsatisfactory. It also highlights the necessity of actively guiding villagers’ public participation .

(2)The second to fourth paragraphs emphasize the importance of the ongoing reform of collective property rights system in China, as well as the focus and limitations of existing research.

(3)Based on the above analysis, the fifth and sixth paragraphs propose the research question of this study: Can the reform of the collective property rights system effectively increase villagers' public participation? Additionally, incorporating other suggestions for revision, we have also included the conclusions and contributions of this study.

These revisions aim to make the opening paragraph clearer, logically more coherent, and relevant to international audiences and the specific context of China. Thank you again for your insightful comments.

The following are the revised parts.

1. Introduction

Public participation, serving as a pivotal force in driving social progress and democratization, has garnered widespread acknowledgment for its significance[1,2]. Various types of public participation, such as voting, engaging in public decision-making, negotiating and discussing and providing volunteer services[3–5], not only effectively convey the needs of the public and ensure that policymaking is scientific and reasonable, but also stimulate the vitality and creativity of various sectors of society, thereby earning widespread attention and research. For instance, public participation can facilitate the sustainable development of the environment, local urban planning, the formulation of transportation policies, and numerous other fields. [6–10], while public apathy will be a critical barrier to public affairs[11]. However, despite the immense potential and value demonstrated by public participation in practice, its effectiveness faces challenges in some rural areas. For example, the old rural economy of farming in the United States presents conservative public participation[12]. Of course, this has also been confirmed by rural areas in developing countries such as Cameroon and Vietnam[13,14]. In rural China, rapid economic development and substantial migrant for work have led to profound changes in the social structure [15,16]. Families and individuals are gradually detaching from the village collective, tending towards individualization, leading to issues such as loose management of rural public affairs and indifference in interpersonal relationships[17]. These problems have further exacerbated the difficulty of democratic governance in rural society, posing unprecedented challenges. Therefore, how to actively guide villagers to participate in village public affairs has become particularly important. This will not only enhance the efficiency of public participation in rural China, but also have far-reaching implications for promoting comprehensive social progress in other countries facing similar issues.

The ongoing rural collective property rights system reform in China represents the fourth major reform practice and innovation after land reform, collectivization, and household contract responsibility system[18,19]. Since its implementation in 2014[20], this reform has set the goal of “establishing a well-defined, fully empowered, smoothly transferable, and strictly protected rural collective property rights system with Chinese characteristics, and protecting and developing the legitimate rights and interests of villagers as members of rural collective economic organizations” [21] . By establishing a clear collective property rights system and operating collective assets in a market-oriented manner, a large amount of idle rural collective resource elements can be activated[22]. From 2014 to 2021, after five batches of pilot projects for this reform, the national inventory and verification of collective book assets amounted to 7.7 trillion yuan, including 3.5 trillion yuan of operating assets [23]. 

The overall progress of the reform has been smooth. Many scholars have explored from the perspective of economic benefits and pointed out that, from a micro perspective, the reform has enhanced the vitality of collective economic organizations, promoted collective income generation and increased villagers' income. From a macro perspective, the reform has boosted productivity and further activated new driving forces for rural development in China[19,24].

It is noteworthy that the governance effects of the reform of the rural collective property rights system have also begun to attract attention. This reform focuses on and revolves around the shareholding cooperative system, which entails quantifying and allocating collective operational assets to individuals and households on the basis of asset inventory and membership determination. Consequently, the reform has facilitated the reconstruction of power relations in villages, empowering villagers with more political and economic rights, enhancing their awareness and capacity to engage in rural public affairs, and enriching the effective forms of deliberative democracy in villages[25,26]. A study, utilizing survey data from 1657 villagers in 87 villages across 18 provinces in China in 2018, analyzes the impact of this reform on village democracy and find that the reform strengthens rural democracy and significantly promotes democratic governance in villages [27]. However, this study also has its limitations, as the independent variable of the reform of the collective property rights system is measured by villagers' responses to whether the reform has been completed. 

Therefore, based on the actual situation in rural China, this study aims to more accurately measure the reform as well as villagers’ public participation, in order to conduct better empirical analysis. The findings of this study reveal that the reform of the rural collective property rights system has significantly promoted the public participation of villagers, especially in terms of total participation, interest expression, and election voting. However, its impact on volunteer service is limited. Mechanism analysis reveals that reform promotes villagers’ public participation by reducing migration for work. Heterogeneity analysis emphasizes the reform has greater impact on suburban villages and villages with better collective economy. Further analysis indicates villagers have heard that reform can significantly increase their public participation. Suggestions of this study include further deepening the reform of rural collective property rights system, promoting local employment and entrepreneurship, implementing the reform differently based on actual village conditions and increasing the promotion and guidance efforts for the reform.

The contributions of this study are threefold. Firstly, use more accurate indicators to measure the reform of rural collective property rights system and public participation. For the reform of the rural collective property rights system, this study measures whether the reform has been completed based on direct question from village level questionnaires, while previous studies verified the completion of the reform through villagers' responses; for public participation, this study establishes three main indicators and one comprehensive indicator, namely interest expression type, election voting type, volunteer service type and total participation. In comparison, previous studies mostly used only one variable for measurement. Secondly, the multi-level linear regression model and other econometric analysis methods used in this study scientifically and comprehensively investigated the impact of the reform on villagers' public participation, as well as the mechanism and heterogeneity. Thirdly, in terms of data, this study used data from 284 village committee questionnaires and 7451 village questionnaires from 10 provinces in China, which is also authoritative and representative. Above of these contributions help proposing countermeasures and suggestions on how to better promote the public participation of villagers in the process of deepening reform.

Comment 2: The literature review and research hypothesis of this article are put together, and this arrangement does not seem to be logical. Generally, the literature review is carried out in the introduction part, or the literature review is delayed. The hypothesis is put together with the theoretical mechanism, and the second paragraph of the current article should be more appropriate as a theoretical mechanism and research hypothesis.

Reply: Thank you very much for providing detailed advice and guidance. The title of the second part does not match its content well and needs to be revised. We have changed the title to “Theoretical analysis and research hypothesis” and made modifications in this part. 

The following are the revised parts.

2. Theoretical analysis and research hypothesis

The reform of the rural collective property rights system encompasses three key aspects: strengthen the management of rural collective assets, carry out property rights reform of operational assets, exploring the forms of realizing rural collective economy (Fig 1)[21]. Throughout the foundational work of this reform, the principle of respecting villagers’ will is paramount. This principle entails harnessing villagers as the primary agents, fostering their innovation and creativity, granting them the right to choose, and ensuring their rights to information, participation, expression, and supervision. Ultimately, this transforms villagers into active participants and beneficiaries of the reform.

Consequently, the reform of the rural collective property rights system will affect villagers’ public participation in three ways: firstly, by strengthening the management of rural collective assets through various measures, it establishes a foundation for villagers to engage in public participation. Experiences from other developing countries indicate that such approaches not only elevate villagers’ awareness and trust in collective assets but also ignite their enthusiasm for participating in public affairs [28,29]. Secondly, promoting the rural collective management asset shareholding cooperative system guarantees villagers’ shareholding rights as members of collective economic organizations[30]. This enables villagers to participate more directly in the decision-making and operations of the collective economy, thereby expanding the channels and depth of their public participation. Lastly, exploring effective collective economy models, encouraging villager innovation, and supporting the standardized transfers and transactions of rural property rights not only promote the diversified development of rural economy, but also offer more platforms and opportunities for villagers’ public participation [31,32]. By ensuring villagers’ right to know, participate, express and supervise in the reform process, the transformation from villagers being mere observers to active participants and beneficiaries of the reform can be truly realized.

Based on the above analysis, this study presents research hypothesis 1:

Hypothesis 1(H1): The reform of rural collective property rights system can promote the villagers’ public participation.

The positive reshaping of rural social structures through the rural collective property rights system cannot be overlooked. Research shows that the reform can effectively attract villagers who work outside to return home[33]. Behind this trend lies villagers’ recognition and expectation of the development potential of their hometown, as well as their strong pursuit of personal rights and a sense of belonging. Returning villagers not only bring rich work experience and entrepreneurial enthusiasm, but also inject new vitality into rural communities, alleviating the issue of rural hollowing and providing a solid foundation for broader public participation among villagers[34,35]. In the process of public participation, villagers can not only express their own demands and interests more directly, but also jointly promote the harmony and development of rural society through collective action, achieving a win-win situation for both individuals and the collective.

Therefore, the hypothesis 2 can be proposed:

Hypothesis 2(H2): The reform of rural collective property rights system reduces the possibility of migration for work, thereby promoting villagers’ public participation.

The impact of collective property rights system reform on villagers’ public participation will be influenced by factors such as the geographical location and economic development level of the village. Firstly, based on the distance from the city center, rural areas can be divided into suburban villages and general villages[36]. Compared with general villages, suburban villages have their own development and governance characteristics. In terms of development, suburban villages have access to more resources and are more likely to take the lead in achieving development goals[37,38]. In terms of governance, suburban villages are often regarded as complex governance units with intertwined interests and mixed personnel, making them more prone to conflicts[39]. But it has also received more attention and policy tilt. The reform of the collective property rights system emphasizes the need for suburban villages to actively establish and improve rural collective economic organizations, and promote the reform of the shareholding cooperative system for operating assets.

Similarly, previous studies have shown that well-developed collective economy not only provide a more solid material foundation and impetus for rural areas, but also greatly enhance villagers’ profound understanding and strong sense of belonging towards the value of collective assets[40,41]. The deepening of this sense of belonging and cognition has laid a solid mass foundation for the smooth progress of rural collective property rights system reform. In this context, with the deepening of the reform of the collective property rights system, it is possible to more accurately and effectively clarify property rights relationships, effectively protect the legitimate rights and interests of villagers, and greatly stimulate their enthusiasm and initiative to public participation.

Thus, it is safe to propose the following hypotheses:

Hypothesis 3(H3): Compared to general villages, the reform of the collective property rights system in suburban villages has a more significant promoting effect on villagers’ public participation.

Hypothesis 4(H4): The better the rural collective economy, the more obvious the promotion effect of the reform of the collective property rights system on villagers’ public participation.

Comment 3: In the empirical part, the current empirical part lacks theoretical depth. It is recommended to combine the hypotheses in the previous article for a more in-depth analysis to increase the theoretical depth.

Reply: Thank you very much for your insightful and constructive comment. We have revised the empirical part according your suggestion. In the revised version, we have combined the empirical analysis results with the analysis in “2. Theoretical analysis and research hypothesis” for a more detailed explanatio

---

## [Decision Letter · Decision Letter 1]

18 Dec 2024

The impact of the reform of rural collective property rights system on villagers' public participation: An empirical study based on CRRS 2020 data

PONE-D-24-33936R1

Dear Dr. Li,

We’re pleased to inform you that your manuscript has been judged scientifically suitable for publication and will be formally accepted for publication once it meets all outstanding technical requirements.

Kind regards,

Dingde Xu

Academic Editor

PLOS ONE

Additional Editor Comments (optional):

After carefully reading the reviewer's comments and the author's responses, I think the author has handled the reviewer's suggestions well. It is suggested to be accepted. Please pay attention to the minor revisions mentioned by the second round of reviewers when proofreading the form.

Reviewers' comments:

Reviewer's Responses to Questions

**Comments to the Author**

1. If the authors have adequately addressed your comments raised in a previous round of review and you feel that this manuscript is now acceptable for publication, you may indicate that here to bypass the “Comments to the Author” section, enter your conflict of interest statement in the “Confidential to Editor” section, and submit your "Accept" recommendation.

Reviewer #2: (No Response)

2. Is the manuscript technically sound, and do the data support the conclusions?

Reviewer #2: (No Response)

3. Has the statistical analysis been performed appropriately and rigorously? 

Reviewer #2: (No Response)

4. Have the authors made all data underlying the findings in their manuscript fully available?

Reviewer #2: (No Response)

5. Is the manuscript presented in an intelligible fashion and written in standard English?

Reviewer #2: (No Response)

6. Review Comments to the Author

Reviewer #2: The authors have responded positively to almost all of the reviewers' comments, and the quality of the article has improved significantly. However, it would have been better if the authors had rechecked the application of the three-line tables in the article and ensured that the three-line tables throughout the article were in line with the specification.

7. PLOS authors have the option to publish the peer review history of their article (what does this mean?). If published, this will include your full peer review and any attached files.

Reviewer #2: **Yes: **Hui Xiao

---

## [Editor Report · Acceptance letter]

15 Jan 2025

PONE-D-24-33936R1 

PLOS ONE

Dear Dr. Li, 

I'm pleased to inform you that your manuscript has been deemed suitable for publication in PLOS ONE. Congratulations! Your manuscript is now being handed over to our production team.

Kind regards, 

on behalf of

Dr. Dingde Xu 

Academic Editor

PLOS ONE